

# Assessing the interaction between mountain forests and natural hazards at Nevados de Chillán, Chile, and its implications for Ecosystem-based Disaster Risk Reduction

Alejandro Casteller[1, 2], Thomas Häfelfinger[1], Erika Cortés Donoso[3], Karen Podvin[3], Dominik Kulakowski[4], Peter Bebi[1]

[1] WSL Institute for Snow and Avalanche Research SLF, Davos Dorf, Switzerland

[2] Instituto Argentino de Nivología, Glaciología y Ciencias Ambientales IANIGLA, CCT-CONICET-Mendoza, Mendoza, Argentina

[3] IUCN, International Union for Conservation of Nature– Regional Office for South America, Quito, Ecuador

[4] Graduate School of Geography, Clark University, Massachusetts, USA

*Correspondence to:* Alejandro Casteller (acasteller@hotmail.com)

**Keywords**: dendrochronology, Ecosystem-based Disaster Risk Reduction (Eco-DRR), natural hazards, protection forest

**Abstract.** Gravitational natural hazards such as snow avalanches, rockfalls, shallow landslides and volcanic activity represent a risk factor for mountain communities around the world. In particular where documentary records about these processes are rare, decisions on risk management and land-use planning have to be based on a variety of other sources including vegetation and tree-ring data and natural hazard process models. We used a combination of these methods in order to evaluate dynamics of snow avalanches and other natural hazards at Valle de las Trancas, in the Biobío Region in Chile. Along this valley, natural hazards threaten not only the local human population, but also the numerous tourists attracted by outdoor recreational activities. Given the regional scarcity of documentary records, tree-ring methods were applied in order to reconstruct the local history of snow avalanches and debris flow events, which are the more important weather-related processes at respective tracks. A recent version of the model Rapid Mass MovementS (RAMMS), that includes influences of forest structure, was used to calculate different avalanche parameters such as runout distances and maximum pressures, taking into consideration the presence/absence of forest along the tracks as well as different modelled return periods. Our results show that local Nothofagus broadleaved forests contribute to a reduction of avalanche runout distances as well as impact pressures on present infrastructure, thus constituting a valuable ecosystem disaster risk reduction measure that can substitute or complement other traditional measures such as sheds.

## 1. Introduction

In mountainous areas, settlements and infrastructure are commonly endangered by natural hazards such as snow avalanches, rockfalls, shallow landslides or volcanic activities. The documentation of past events, including a detailed description of their properties (e.g., magnitude, affected area, type of damage) as well as their effects on and interactions with ecosystems are important for assessing risk (Stoffel and Huggel, 2012; Gądek et al., 2017). In mountain ranges with a long history of settlement such as the European Alps, documentary records of gravitational natural hazards have been compiled for more than five centuries due to the close contact between mountain inhabitants and such phenomena (Laely, 1984). Yet, only more contemporary records have provided





suitable information for accurate risk assessments (Wisner, 2004). In many other mountainous regions, human settlement has been rather limited until the 20th century and has substantially increased only during the last decades (Riebsame et al., 1996; Romero and Ordenes, 2004; Su et al., 2011). It is thus particularly important in such more remote mountain areas, in which human populations and infrastructure are expanding, to learn more

about different natural hazard processes and how they interact with mountain ecosystems.

In regions where documentary records of past events of gravitational hazards are limited, dendrochronological methods provide a valuable and accurate tool for their spatio-temporal reconstruction (Braam et al., 1987; Shroder, 1980; Pop et al., 2016). The maximum length of this type of reconstruction depends primarily on the maximum age of the trees available for sampling at the study area. These ages, in turn, depend on two main

factors: (i) the site history related to human interventions such as induced forest fires, logging and grazing (Mundo et al., 2013) and (ii) the frequency of disturbance events including avalanches and mass movements (i.e., more frequent events typically result in lower tree ages; Corominas and Moya, 2010). Documented damages produced by gravitational natural hazards (in particular snow avalanches) to old buildings, such as churches, has made it possible to couple long-term dendrochronological reconstructions with very precise

information on partial and total destruction of infrastructure, thus validating the use of tree-ring methods for such studies (Fliri, 1998).

Forests can protect infrastructure from natural hazards in mountainous areas (Brang et al., 2006; Getzner et al., 2017). Mountain forests are known for stabilizing the snowpack in potential avalanche release areas and thus help preventing avalanche initiation (Weir, 2002; Bebi et al., 2009,). In the case of small to medium-scale snow

avalanches (i.e., < ca. 10000 m$^3$), forests can also extract snow from the avalanche flow through deposition on the upslope-facing part of tree stems and on other obstacles (Feistl et al., 2014). Forest parameters such as stand composition, stem density and terrain roughness have thus not only an effect on stabilizing the snowpack in potential release areas (Bebi et al., 2009; Viglietti et al., 2010), but may also influence the spatial extent of small and medium-sized avalanches (Teich et al., 2012). However, snow avalanches can break, uproot and overturn

trees (de Quervain, 1978; Takeuchi et al., 2011). Especially for large avalanches (i.e., > 10000 m$^3$) that start far above the treeline, forests have therefore only a marginal influence on the kinetic energy and runout extent of avalanches (Bartelt and Stöckli, 2001). Nevertheless, mountain forests have thus been recognized as a cost-efficient but spatially and temporally varying protection Eco-DRR measure (Olschewski et al., 2012; Renaud et al., 2013). Eco-DRR has been defined as "the sustainable management, conservation and restoration of

ecosystems to reduce disaster risk, with the aim of achieving sustainable and resilience development" (Estrella and Saalismaa, 2013).

In the Southern Andes, human settlement in villages and cities as we know them nowadays started only about 150 years ago, coinciding with one of the largest European immigration waves and the expansion of populations towards the slopes of the Andes (Germani, 1966). This process resulted in a sudden exposure of human

populations and infrastructure to relatively well-known natural hazards but in a new environment. Since then and up to the present, knowledge transfer has been made from (particularly) the European Alps to the Southern Andes to better understand the triggers, dynamics and resulting hazard of such natural hazards (Casteller et al., 2008; Stoffel et al., 2012). Yet, further major efforts are still needed to fulfil these goals.

Native forests in Chile provide several services that directly or indirectly benefit human wellbeing, including the

provision of food and non-timer products, regulation of the water supply, capture of carbon, conservation of soil,





as well as the provision of habitat for biodiversity and space for recreation and tourism (Lara et al., 2003). These services also reduce the underlying vulnerabilities generated by the socioeconomic systems. The expansion of farming and forestry activities and the exploitation of hydropower resources have led to a fragmentation of ecosystems within the study area. Increasing urbanization associated with growing tourism in the area has

expanded the infrastructure and housing, putting more pressure on the local landscape and its biodiversity (Cordero et al., 2014) and increasing the exposure of people and infrastructure to snow avalanches and other natural hazards.

The overreaching goal of this investigation was to evaluate the role of mountain forest ecosystems as an Eco-DRR measure against natural hazards at Valle de Las Trancas, Nevados de Chillán, in the Biobío Region of

Chile. Within this region, avalanche-related casualties have been reported from the ski resort Nevados de Chillán and at the road connecting the village of Las Trancas with infrastructure located along the valley ending at the ski resort (La Discusión, 2011). Yet, the risk of snow avalanches in this area is largely unknown and, besides road temporal closures, no other passive or active measures (except within the limits of the ski resort Nevados de Chillán) are in place to reduce avalanche-related risks.

Besides snow avalanches, also other natural hazard processes such as volcanic activity, rockfall, debris flows and shallow landslides affect our study area. We thus complemented our detailed study on snow avalanche and vegetation interactions with investigations on other natural processes that occur in the area. This allowed us to evaluate the importance of native forests in relation to different natural hazards in the study area and to provide a suitable basis for proposing adequate Eco-DRR measures.

This study is part of the EPIC (Ecosystems Protecting Infrastructure and Communities) project, which has been implemented by the International Union for Conservation of Nature and several local partners and aims at demonstrating with different study cases around the world how improved ecosystem management can reduce the risk of disasters and help to adapt to climate change. The study is focused answering the following questions: (i) what is the effect of avalanches and debris flows on the structure of affected forests?; (ii) what is the influence of

forest on avalanche runout and avalanche hazard?; (iii) What assessment approach is most suitable to assess avalanche-forest interactions where information on past disturbance events is limited?; and (iv) what are potential Eco-DRR measures to reduce overall risks in the study area?

## 2. Study area

The study area is located in the Biobío Region of Chile in northern Patagonia. We focused our investigations at

Valle de las Trancas, an east-west facing valley that is located within the limits of the transition zone of the Biosphere Reserve Corredor Biológico Nevados de Chillán– Laguna del Laja (Fig. 1). This Biosphere Reserve was declared as such by UNESCO on 29 June 2011. The presence of the northernmost population of the Patagonian huemul (Hippocamelus bisulcus), categorized as endangered by IUCN's red list of threatened species, and other rare species, as well as the high genetic diversity, are key issues that have made this Biosphere

Reserve a high priority for conservation (San Martín, 2014).

In recent decades Valle de las Trancas has experienced high economic development due to its popular tourist attractions, mainly related to mountain sports and thermal springs (Pfanzelt et al., 2008). The municipality of Pinto, in which the Valle de Las Trancas is located, lacks official land-use plans and regulations, leading to an



uncontrolled establishment of new buildings in the area. This development has also increased the demand of water for local consumption and of wood for heating.

A growing demand along Valle de Las Trancas of new tourist infrastructure, in an area lacking appropriate hazard maps, can result in an increased exposure to gravitational processes with serious risk for human populations. In addition, the ski resort Nevados de Chillán, located in the basin headwater, is one important local actor of the private sector authorized to cut patches of native forests to clear the land for new ski tracks. This potentially results in trade-offs with the forests' provision of protection against snow avalanches.

### 2.1. Biogeographic setting

The investigated area is located between the temperate and Mediterranean macrobioclimates (Luebert and Pliscoff, 2006). It is characterized by dry summers and cold winters; more than 75 % of the annual precipitation fall in winter between April and September (Donoso, 1993). Local climate data obtained from a weather station located at Las Trancas (36°54'41" S, 71°28'44" W), just at the entrance of homonymous valley, report that the mean annual precipitation between 1998 and 2012 was 1007 mm. Mean annual temperature during the corresponding time period was 13.2 °C. Longer reliable climatic records, as well as specific data on snow heights and extreme snowfall events, are unavailable for the study area.

A high diversity of geological substrates, caused by high volcanic activity, characterizes the study region. Adjacent to the study area is located the Nevados de Chillán volcano, which consists of the Cerro Blanco and Las Termas subcomplex (formed about 40000 years ago). The most recent eruptions from the Nevados de Chillán volcano occurred in the years 1973, 2003 and 2016; due to the exposure of tourist facilities, it is categorised as one of the volcanos with the highest damage potential in Chile (Dixon et al., 1999). These diverse geological substrates are overlaid with characteristic soils classified as inceptisols, which are developed out of these volcanic products (Freiberg, 1984). The high diversity of geological substrates leads in turn to 11 different vegetation units (Pflanzelt et al., 2008). Close to the riverbed in humid localities, Nothofagus dombeyi (Mirb.) Oerst. forests can be found. The other parts of the forest are mixed forest stands of Nothofagus dombeyi, Nothofagus pumilio (Poepp. & Endl.) Krasser and Nothofagus obliqua (Mirb.) Oerst. In sections of the study area, Chusquea culeou-Coirones assemblage, with the dominant representative species Chusquea culeou Desvaux, can be found. On the upper portion of the slopes near the ridges, Nothofagus pumillio krummholz-type vegetation complements the forest diversity.

### 3. Material and methods

### 3.1. Remote sensing analyses

An identification of gravitational natural hazards within the study area was conducted in an early stage of this investigation and complemented with geographical information subsequently obtained. We firstly focussed on spatially demarcating the avalanche tracks that reached (according to available records) infrastructure along the road of Valle de Las Trancas. Spatial analysis and interpretation of aerial images was made using ArcMap 10.2 (ESRI, 2013). Complementary, Google Earth images were used to display the information and as a basis for the interpretation of aerial images. Additionally, available aerial images of the study area (from years 1978, 1979,





1982, 1998 and 2000) were stereoscopically analysed in order to identify chronological modifications in the forest structure due to natural disturbances following the approach of Bebi et al. (2001) and Panayotov et al. (2011). For the assessment of the current forest structure in selected avalanche tracks, we complemented these analyses with a high-resolution remote sensing data (including a Digital Terrain Model [DTM] and cover image) obtained in February 2015 with an eBee drone (resolution: 8 cm/pixel).

We complemented these data with repeated field surveys. Trees that were either sampled or simply identified for features of interest were individually positioned using a portable navigation system (model: GPSMAP® 64s). Forests in which the spatial extent of past avalanches was visible in the field were mapped using the same method, in particular borders and avalanche runouts. Photographs were taken of avalanche release zones (which are mostly above the tree-line and on steep slopes) from different angles and locations (Fig. 2). These photographs subsequently helped for spatial demarcation of these release zones using a Geographic Information System (GIS).

Between the village of Las Trancas and the base of the ski resort Nevados de Chillán (36°54'12" S, 71°24'32" W; i.e., end of the road) we were able to identify 18 active (i.e., with at least one documented event) avalanche tracks (Fig. 3). In some of the avalanche tracks, we also observed abundant evidence of the occurrence of debris flows which were confirmed in some cases by documentary records.

### 3.2. Forest structural and dendrochronological analyses

Two different approaches were applied to reconstruct spatio-temporal occurrence of gravitational natural hazards. The first one focused on identifying particular trees at selected avalanche tracks with the largest amount of visible damage (which is typically associated with repeated events), such as broken stems and/or branches, decapitation, uprooting and exposed scars. From the total of 18 avalanche tracks identified, we applied this approach (called hereafter "selective sampling") in two of them (defined as av6 and av9; Table 1). We selected these two tracks for the following reasons: av6 had one recorded avalanche event in 2011 (that reportedly caused one casualty) that could be used for calibration purposes, whereas av9 presented a very clear avalanche runout zone and borders with numerous trees evidencing recent disturbances. Selected trees were sampled using increment borers (Haglöfs; width 0.5 cm) as well as hand saws, looking in all cases to avoid unnecessary damage to living trees in this protection forest (Bebi et al., 2001; Dorren et al., 2004). Morphological features of sampled trees, such as height of the damage along the main stem, size of the scars and inclination of the stems were registered, complemented by drawings and pictures.

The second approach aimed at evaluating the forest age and structure in specific plots. The location of these forest plots was selected according to the following criteria: (i) placing plots at each disturbance patch (i.e., avalanche/debris flows track) with the presence of forest; and (ii) at each forest stand in between two disturbance patches. Each plot had a radius of 10 m, thus covering a surface of 314 m$^2$. Parameters registered at the forest plots included terrain roughness, canopy density, properties of the shrub layer and slope angle. For each tree within the plots with a diameter at breast height (DBH) larger than 8 cm, further parameters were recorded: species, diameter, height, and social status. Additionally, at least 5 randomly-selected trees were cored at each plot to estimate the age of the stands. The gathered data from the forest structure plots were analysed using the statistic software R (R Development Core Team, 2015).



All tree-ring samples were prepared following standard dendrochronological techniques (Stokes and Smiley, 1968) and subsequently ring-width measured using a LINTAB system from RINNTECH, Germany. The software COFECHA was utilized to detect (and if necessary to correct) dating errors, using as a reference chronology tree samples from adjacent undisturbed sectors of the forest (Holmes, 1983). Further, every tree-ring sample was carefully examined for the presence of scars, growth releases and growth decreases, as well as peculiar features in the wood such as frozen rings or wood compartmentalization. Growth reactions were grouped in three different categories following Dubé et al. (2004). Both the expert (Stoffel and Corona, 2014) and indices (Shroder, 1978) approaches were then applied in a comparative way to achieve a final reconstruction of disturbance events at each avalanche/debris flow track.

In regions where the precipitation patterns determine the occurrence of debris flows during the vegetation period of trees, it is possible to differentiate between the occurrence of snow avalanches and debris flows by identifying the position of the injuries within the tree rings (i.e., dormancy for avalanches and during active growth for debris flows; Stoffel et al., 2012). This is not the case at our study area, where most precipitation events take place in winter and thus it was not possible to differentiate between the two types of disturbances by means of tree-ring analyses.

### 3.3. Snow avalanche simulations

Numerical simulation models have been used to reconstruct snow avalanches in various mountain regions including the Andes (Valero et al., 2016), Central Europe (Blahut et al., 2017), Japan (Takeuchi et al., 2011) and Scandinavia (Gauer and Christiansen, 2016). The software RAMMS was used to conduct snow avalanche simulations at Valle de las Trancas. RAMMS is a two-dimensional software to calculate mass movement dynamics in a three-dimensional terrain (Christen et al., 2010). We applied RAMMS: AVALANCHE version 1.6.25 ©WSL/SLF with its new forest applications. Compared to standard simulations with RAMMS::AVALANCHE, this version differentiates among the effects of different forest structural characteristics on avalanche runout. Therefore, spatial information on crown coverage, roughness, and DBH had to be defined from the forest plot survey and remote sensing data for the simulations at each track. Based on the forest type, crown coverage and roughness in the area, we subsequently deduced spatial explicit information on the detrainment coefficient K [kg m$^{-1}$ s$^{-2}$], which represents the breaking power that a forest exerts on the avalanche flow (Feistl et al., 2015). Information regarding snow height and snow density along the track and at the release areas was not available for each avalanche event to be considered in our simulations and thus had to be estimated based on typical and extreme snow situations for this region. As the exact location and size of avalanche releases was also not available for all events, we conducted GIS-analyses using topographic parameters (e.g., slope, aspect, confinement and distance to the next ridge) in order to best estimate this information (Maggioni and Gruber, 2003; Bühler et al., 2013). Simulations with both ca. 10 and ca. 100-year return periods were conducted. For extreme scenarios with ca. 100-year return periods, we considered also the largest possible release areas based on topographical settings in and around potential release areas. Avalanche friction values were differentiated between open, channelled, and gully slopes and also for different forest structural characteristics following the approach and protocol of Feistl et al. (2015) and Feistl et al. (2016).





At avalanche tracks 6 and 9 (av6 and av9, respectively), specific information regarding runout distances from past events was available (both from tree-ring data and documentary records). Out of the results of our tree-ring analyses, trees with synchronous reactions to avalanche disturbances at these run-out zones were used to delineate the affected area of past (tree-ring dated) events. We then compared this information with the spatial data provided by the numerical simulations with RAMMS. To further evaluate the effect of local forests on avalanche dynamics, different simulations considering both the presence and absence of forest were conducted. Further, avalanche simulations were conducted at a selected track (av6) in order to quantify the improved protective capacity of the forest by conducting additional afforestations, with the assumption that the additionally-considered forests have similar functional properties as the current forest.

## 4. Results

### 4.1. Forest structure in disturbed and control plots

Our analyses show that avalanches and debris flows play a role in influencing the structure of the forest at the study area. Canopy densities of the investigated plots range between 20 % and 90 % (average value: 60 %; Fig. 4). By grouping plots in different categories (i.e., no disturbance, debris flows and snow avalanches), we observe that avalanche disturbances reduce the canopy density by about 20 % in relation to undisturbed forest, whereas for debris flow this reduction is 10 % (Fig. 5); these results are however not significant ($p$-value > 0.05).

### 4.2. Reconstruction of events based on tree-ring analyses

The results of our tree-ring analyses allowed us to reconstruct past snow avalanche and debris flows years at the investigated tracks. Scars and abrupt growth changes provided the most reliable indicators for past disturbance events. Table 2 contains a list of all years during which, according to our reconstruction, either snow avalanches or debris flows occurred at the investigated tracks. Only forest plots located in disturbance patches are considered here (see Fig. 3 for reference), in addition to tracks av6 and av9 in which the selective sampling was conducted.

The first year during which an avalanche event was detected in our tree-ring data is 1942. The main factors that prevent the extension of our reconstruction beyond that year are, on the one hand, the young age of the local forests and, on the other hand, wood decay. At avalanche track 6 (av6; using selective sampling), a forest plot (fp8) was also selected in order to compare the event reconstruction using both approaches. We observed that only two events are captured at fp8 since 1942, whereas the selective sampling allowed the reconstruction of 23 events (Table 2). Noteworthy, during 1995 and 2000 a particularly high occurrence of events was registered at many of the investigated tracks.

By making a linkage between trees showing synchronic avalanche reactions we could reconstruct five different runouts at av6 and four of them correspond to more than one year (Fig. 6). Events with a shorter runout compared to the runout of 2011 could not be reconstructed due to the absence of witnessing trees.





### 4.3. Influence of forest on avalanche dynamics

In order to evaluate the influence of the Nothofagus-dominated forests on avalanche dynamics (and thus quantify its protective capacity), avalanche simulations considering both the presence (in its actual situation) and absence of forest were conducted. Simulations for avalanche track 9 (av9) are shown here as an example. This track is largely forested in its runout zone providing a potentially important protective function for the road crossing the valley. Our simulations show that for a 10-year return period scenario without considering forests, the spatial extent of avalanches is 19 % larger as compared to avalanches running in a forested track. The runout of the avalanche in this not-forested scenario thus reaches the road, in contrast to the runout of the forested scenario, which does not reach the road or other infrastructure (Fig. 7). Considering a 100-year return period scenario for the same track, the absence of forest would determine that avalanches could reach the road, being their spatial extent 23 % larger than with forests along the track (Fig. 8)

Simulations of avalanches considering additional afforestation scenarios were conducted for av6, which shows the largest potential impact on infrastructure along the road (see also Fig. 6), considering a 10-year return period scenario. Our results show that the runout of avalanches would not be significantly shortened (5 %) due to afforestations at the track, but that there would be a reduction of avalanche maximum pressures at the intersection with the road by 24 % (Fig. 9).

### 5. Discussion

Snow avalanches and landslides are natural disasters that have not been thoroughly studied in Chile (Espinoza et al., 1983). In this investigation we identified the main natural hazards representing a risk for local communities at Valle de las Trancas within the Biosphere Reserve Nevados de Chillán – Laguna del Laja. Besides the continuous growth of its human population, the valley receives a large and increasingly growing amount of visitors attracted by leisure activities and a magnificent mountain landscape throughout the year. Yet, the lack of land-use planning and regulation that considers the occurrence of natural phenomena such as snow avalanches and debris flows, contributes to risk for people and infrastructure. This study assessed the role of native mountain forests, represented here by different species of Nothofagus, as a potential resource to apply an Eco-DRR approach in order to reduce hazard exposure. It is assumed that native mountain forests could be further conserved and sustainably managed to improve their protective capacity against disturbances such as avalanches or landslides, and thus reducing the risk of disasters in the area.

The lack of documentary records of spatio-temporal patterns of gravitational natural hazards in the Southern Andes is a recognized limitation to estimating related risks. Efforts have been made in the past to apply tree-ring methods in this region in order to substitute for or complement documentary records, which, even if available, tend to be fragmentary (e.g., Mundo, 2007; Casteller et al., 2009, 2011, 2015). Methodological limitations exist however: for example, a reconstructed avalanche event year implies the occurrence of at least one event during that particular winter, for which it is not possible to determine if more than one event occurred at the same track during the same winter. In contrast, distinguishing among events occurring in different times of the year is possible by identifying the position of reactions within tree rings: in regions where debris flows occur in summer, it is normally possible to distinguish between their associated tree-ring reactions (from early wood to





late wood) from those originated by snow avalanche impacts (the reaction can be found in this case at the boundary of two rings; Stoffel et al., 2006). In the current investigation area, however, we were not able to distinguish between avalanche and debris flow events using tree-ring methods because both processes occur typically during the same season (winter), when most of the precipitation commonly occurs. However, through available records, types of damages in the forest, topographical features (including channel geometry) and process modelling we were able to distinguish - to a large extent - one process from the other.

As we have shown, different sampling techniques can lead to different results when analysing temporal patterns of snow avalanches and debris flows. The selection of single trees with visible multiple damages of past disturbance events at the borders (and for snow avalanches also at the runouts) of the tracks can provide longer event reconstructions as compared to sampling in plots adjacent to disturbed areas. Yet, sampling plots, as we have shown, can also provide valuable data related to forest structure, from which it is possible to infer past disturbances. In addition, forest plots are a valuable source of information to be analysed with remote-sensing tools and for larger areas. A combination of both sampling approaches was found to be most useful.

The role of mountain forests as protective measures against natural hazards has been acknowledged for centuries in the European Alps, where mountain settlements are in close contact with phenomena such as snow avalanches and landslides (Price and Thompson, 1997; Getzner et al., 2017. The use of the new model RAMMS::Avalanche (that considers forest effects), developed and first tested in the Alps, was used in this investigation to reconstruct past avalanche events and to determine potential avalanche scenarios. Such simulations, with a few exceptions (Casteller et al., 2008), are practically non-existent in the Andes. This type of multi-methodological approach has proven to be most effective (Gądek et al, 2017), in particular in regions where documentary data of past events is scarce. In addition to the methods and data used in this study, the availability of detailed data on e.g. size and precise location of avalanche release zones (including snow depth), type of snow and snow entrainment along the track would allow even more robust event modelling.

Our results indicate that local native broad-leaved forests may influence avalanche dynamics, particularly in small to medium-sized events if the distance between the release area of the avalanche and the upper timberline is less than ca. 200 m (Weir, 2002; Teich et al., 2012). The effect of such broadleaved forests on avalanche releases is certainly less than of evergreen forests due to lower interception rates during snowfall (Lundberg et al., 2004; Bebi et al., 2009). However, their effect on reducing runout distances may still be considerable, in particular for small avalanches that do not cause widespread tree breakages (Feistl. et al., 2014). In such cases, a shortening of runout distances and reducing impact pressures can largely be explained by the extraction of mass and its momentum of the avalanche flow by the trees. These results suggest that appropriate and spatially well-targeted afforestations may potentially reduce risks related to snow avalanches in our study area. In contrast to grey measures such as  sheds (Campbell et al., 2007), afforestations would have clearly less detrimental effects on the provision of other ecosystem services, in particular if they are conducted with regional tree species and locally limited to appropriate site conditions (Schönenberger, 2001). In line with expectations of the local community(Cortés Donoso, 2017) , strict controls of further forest cutting within the Biosphere Reserve are needed in order to allow a sustainable protection against natural hazards without major impacts on the forest structure and ecosystem dynamics (Bebi et al., 2017).



According to gathered data at the study site (Cortés Donoso, 2017), the two most relevant risks for the local community at Valle de las Trancas are snow avalanches and volcanic activity. This large volcanic complex has recently reached over 18 months of visible activity. In addition, the occurrence of forest fires in Chile has increased exponentially during the last years representing a major national hazard (González et al., 2011).. While potential effects of Eco-DRR measures against volcano activity are very limited, such measures represent a window of opportunity in terms of protection against snow avalanches. The promotion of services provided by the native forest and the participation of the local community to conserve and sustainably manage the forest are issues to be closely considered.

## 6. Conclusions

Our study shows that a combination of different methods and approaches is crucial to a comprehensive understanding of interactions between natural hazards, forest ecosystems and human drivers and to provide a sufficient basis for decision support. This is particularly true for complex mountainous regions like the Nevados de Chillán, where historical records on natural hazard regimes are scarce. Overall, the combination of different methods applied in this study suggests that the conservation of regional native forests may contribute, at least in specific parts of the region, to their protective function against natural hazards, while also being crucial for other ecosystem services demanded by the local population. Given the relevance of snow avalanches and other gravitational natural hazards for the local community and for tourism, it is important to better integrate related spatial information as an input for land-use planning tools along the Valle de las Trancas. This data integration can complement already-existing volcanic hazard maps to prevent human casualties and damage to infrastructure.

### Acknowledgements

The authors thank the Ministry of Environment of Chile (MMA) for its political support throughout the implementation of the project and the International Climate Initiative (ICI) of the German Federal Ministry for Environment, Nature Conservation, Building and Nuclear Safety (BMUB) for funding the project. Gabriela Jara Aburto and Gabriel Orozco from SERNAGEOMIN are acknowledged for providing from the beginning until the end of the project with support and valuable information. Field assistants Ernesto Corvalán, Jorge Silva, Eugenia Marcotti, Carolina Córdova, Viola Debus, Álvaro Gutiérrez and Paul Szejner permitted the successful conduction of field activities. Likewise, our gratitude goes to Jaime Soto and Leandro Olivares who conducted drone flights at the study area and shared this information with us for the avalanche modelling. Last but not least, we thank Marc Christen, Yves Buehler and Lukas Stoffel at the SLF Institute for their valuable insight related to the avalanche modelling.

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





Table 1: Samples collected at different tracks and forest plots.

| Code | Sector | Cores (n) | Cross-sections (n) |
|---|---|---|---|
| av6 | avalanche track & runout | 121 | 5 |
| av9 | avalanche track & runout | 73 | 9 |
| fp1 | unaffected forest | 6 | 0 |
| fp2 | unaffected forest | 5 | 0 |
| fp3 | debris flow runout | 5 | 0 |
| fp4 | unaffected forest | 5 | 0 |
| fp5 | debris flow runout | 6 | 0 |
| fp6 | unaffected forest | 5 | 0 |
| fp7 | unaffected forest | 5 | 0 |
| fp8 | avalanche runout | 5 | 0 |
| fp9 | avalanche runout | 5 | 0 |
| fp10 | unaffected forest | 6 | 0 |
| fp11 | unaffected forest | 5 | 0 |
| fp12 | avalanche runout | 4 | 0 |
| fp13 | unaffected forest | 5 | 0 |
| fp14 | unaffected forest | 5 | 0 |
| fp15 | unaffected forest | 6 | 0 |
| fp16 | unaffected forest | 5 | 0 |
| fp17 | unaffected forest | 6 | 0 |
| fp18 | avalanche runout | 6 | 0 |
| fp19 | avalanche runout | 5 | 0 |





Table 2: Reconstructed avalanche and debris flow events at the investigation area. Fp3 and fp5 show event years in the debris flow tracks (df3 and df4, respectively), whereas fp8, av6, fp9, fp12, av9, fp18 and fp19 show event years in different avalanche tracks (av6, av6, av7, av8, av9, av17 and av18, respectively).

| Avalanche/ debris flow years | fp3 (df3) | fp5 (df4) | fp8 (av6) | av6 | fp9 (av7) | fp12 (av8) | av9 | fp18 (av17) | fp19 (av18) |
|---|---|---|---|---|---|---|---|---|---|
| 2014 | | x | | | | | | | |
| 2012 | | | | x | | | x | | |
| 2011 | | | | x | | | x | | |
| 2010 | x | | | | | | x | | |
| 2005 | | | | x | | | | x | |
| 2002 | | | | | | | x | | |
| 2000 | | | | x | | x | x | x | |
| 1995 | x | x | x | x | x | x | x | | |
| 1992 | | | | x | | | | | |
| 1991 | | | | x | | x | x | | |
| 1989 | | | | x | | | | x | x |
| 1986 | | | x | x | | | | x | |
| 1984 | | | | x | | | | | |
| 1983 | | | | x | | | | x | x |
| 1982 | | | | x | | | | | |
| 1980 | | | | x | | | | | |
| 1977 | | | | x | | | | | |
| 1975 | | | | x | | | | | |
| 1974 | | | | x | | | | | |
| 1972 | | | | x | | | | | |
| 1968 | | | | x | | | | | |
| 1964 | | | | | x | | | | |
| 1962 | | | | x | | | | | |
| 1961 | x | | | | | | | | |
| 1957 | x | | | | | | | | |
| 1954 | | | | x | | | | | |
| 1951 | x | | | x | | | | | |
| 1947 | x | | | | | | | | |
| 1945 | | | | x | | | | | |
| 1942 | | | | x | | | | | |



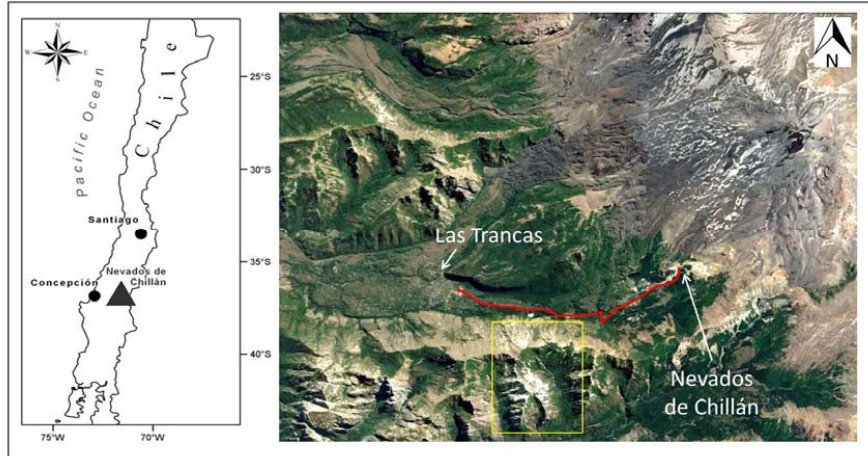

Figure 1: Study area. Along the road connecting Las Trancas with Nevados de Chillán we focussed our studies on natural disturbances and protection forest. The yellow squared zone shows that the north-facing slope is much less vegetated as compared to the south-facing slope, which is mostly explained by the lower radiation received

5 on the latter and thus longer remaining snow cover that provides water in drier periods of the year (see also Fig. 2).



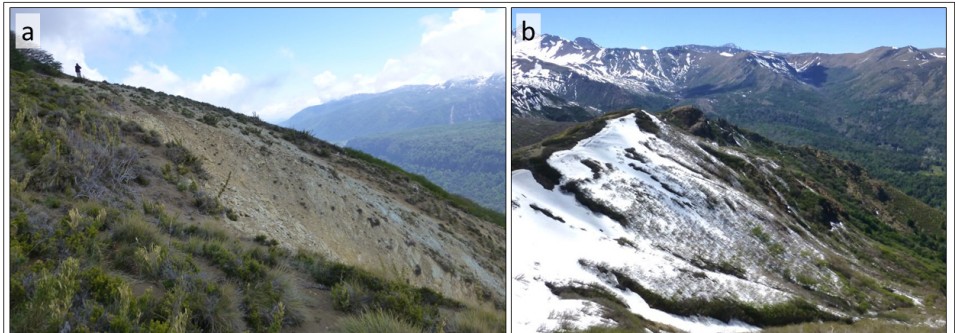

Figure 2: (a) Release area of avalanches, which in this particular track is also the initiation area of debris flows
(df4; see Fig. 3 for reference). Just on the other side of the ridge (b; pictures taken the same day) we can observe
a large amount of snow remaining from the previous winter.



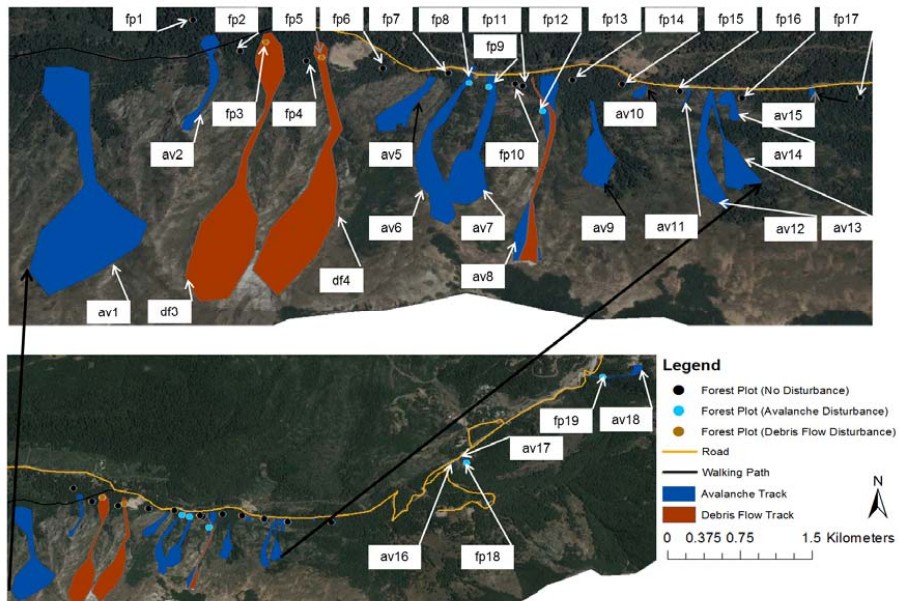

Figure 3: Disturbance tracks and forest plots (fp) at the study area. Tracks depicted in blue correspond to snow avalanches, whereas those in red correspond to both snow avalanches and debris flows. A total of 18 tracks were identified in the study area and 19 forest plots were defined.





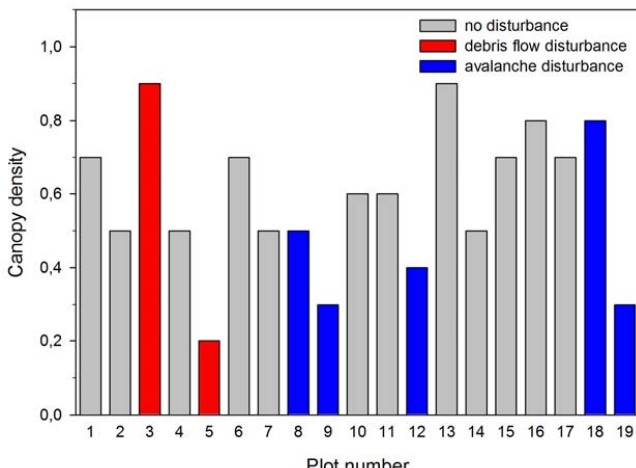

Figure 4: Canopy density distribution (%) per plot.




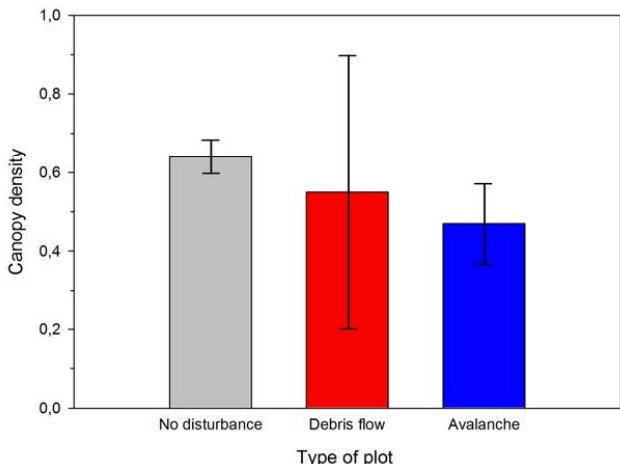

Figure 5: Canopy density distribution (%) per disturbance.





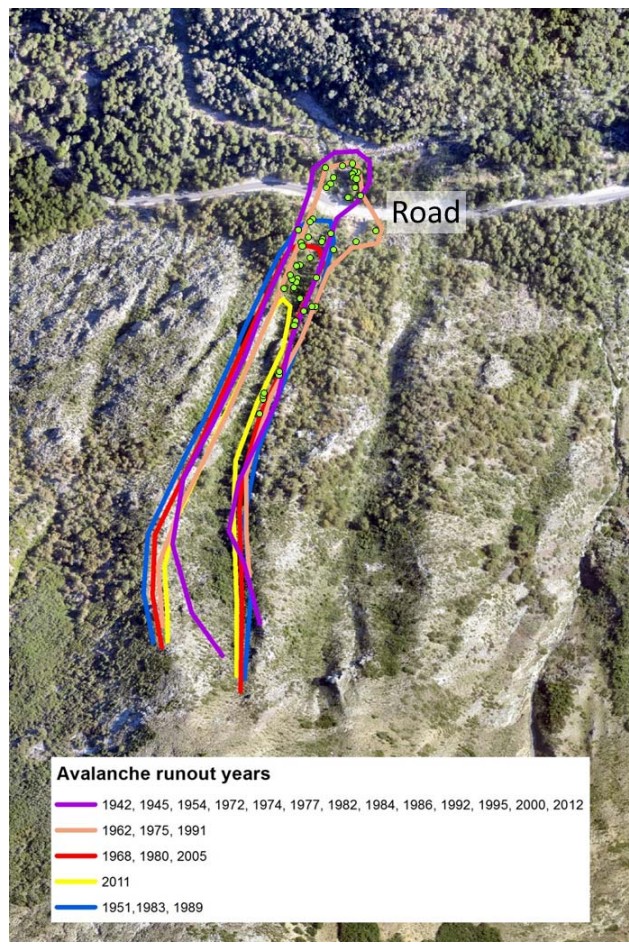

Figure 6: Reconstructed borders and runouts at avalanche track 6 using tree-ring data since the first registered event in 1942 up to the sampling time in 2015.



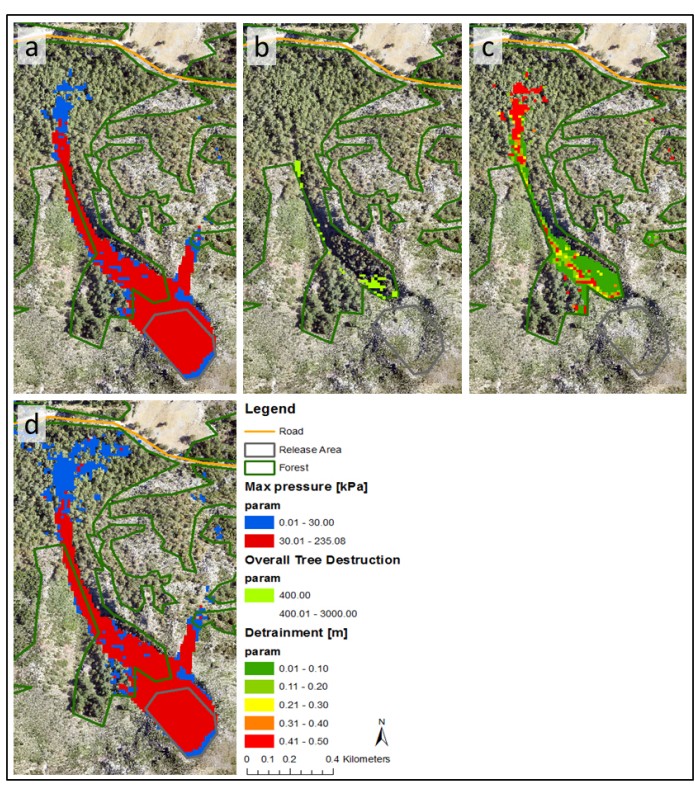

Figure 7: Avalanche simulations at avalanche track 9 with a 10-year return period: (a) maximum pressure of an avalanche scenario with forest; (b) area where forest is destroyed; (c) detrainment in the forest; and (d) the maximum pressure of an avalanche scenario without considering forest.





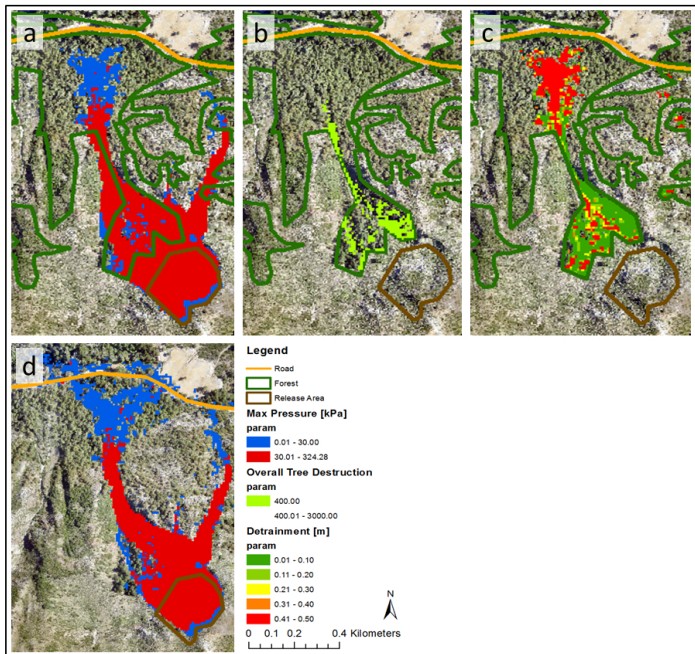

Figure 8: Avalanche simulations at avalanche track 9 with a 100-year return period: (a) maximum pressure of an avalanche scenario with forest; (b) area where forest is destroyed; (c) detrainment in the forest; and (d) the maximum pressure of an avalanche scenario without considering forest.





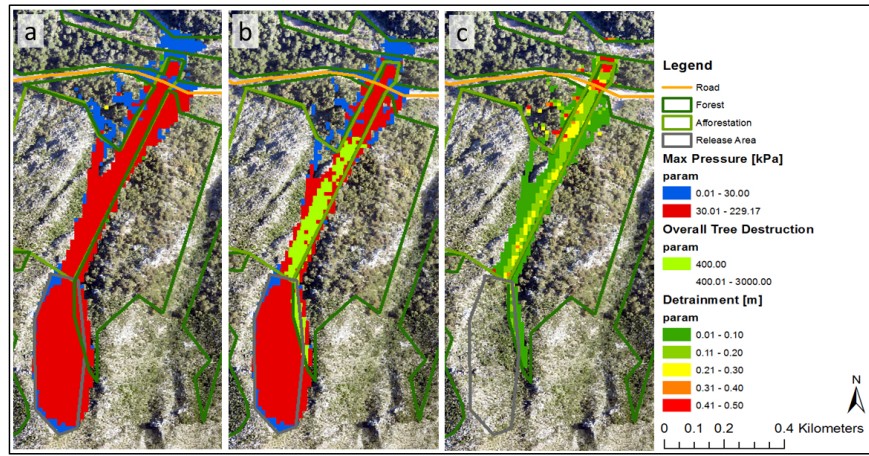

Figure 9: Influence of additional afforestation in avalanche track 6 with a 10-year return period: (a) maximum pressure in an avalanche simulation without additional forests; (b) maximum pressure and overall tree destruction in an avalanche simulation with additional forest; and (c) detrainment in an avalanche simulation with additional forest.