# Peer review of "Assessing the interaction between mountain forests and snow avalanches at Nevados de Chillán, Chile, and its implications for Ecosystem-based Disaster Risk Reduction"

_Natural Hazards and Earth System Sciences, 2017_

## Referee Comment (RC1) · Anonymous Referee #1 · 12 Jan 2018

1. The title is this paper needs to be more detailed: The papers deals with snow avalanches and to a smaller extent also with debris flows

2. The introduction needs a better structure:

The first part deals with event documentation (in addition, the first and second sentence have no link). Then how settlement in mountains other than the Alps increased substantially and "thus it is particularly important in such more remote mountain areas, in which human populations and infrastructure are expanding, to learn more about

different natural hazard processes and how they interact with mountain ecosystems". Why that? I would say it is important to know how these processes potentially affect the humans living there.

Next the introduction continues with how lacking event information can be obtained via dendrochronological methods.

Then we pass on to the statement "forests can protect infrastructure from natural hazards in mountainous areas" and after that the authors only mention the interaction between forests and snow avalanches. and the paragraph ends with Eco-DRR, which is shortly introduced.

The goal is the paper is not specific enough: The overreaching goal of this investigation was to evaluate the role of mountain forest ecosystems as an Eco-DRR measure against natural hazards at Valle de Las Trancas, Nevados de Chillán, in the Biobío Region of Chile. The primary focus of the whole paper is on snow avalanches.

3. Make sure there is a link between the methods chapter and the results chapter. Suggestion for the structure of the methos chapter 1. Introduction - find the relevant tracks in the forest 2. Forest structure analysis 3. Tree ring analysis 4. Avalanche simulation with forest effect

4. The methods chapter mentions (p. 6, line 14): "In some of the avalanche tracks, we also observed abundant evidence of the occurrence of debris flows which were confirmed in some cases by documentary records" and "it was not possible to differentiate between the snow avalanches and debris flows". Then in the results (p.7 /line 18) the authors mentions: "The results of our tree-ring analyses allowed us to reconstruct past snow avalanche and debris flows years" This all confusing. In the discussion again: "In the current investigation area, however, we were not able to distinguish between avalanche and debris flow events using tree-ring methods because both processes occur typically during the same season (winter), when most of the precipitation commonly occurs. However, through available records, types of damages in the forest,

topographical features (including channel geometry) and process modelling we were able to distinguish - to a large extent - one process from the other". The paper needs to provide more clarity!

5. The paper refers to the snow avalanches and debris flows as: natural hazards, natural events, disturbances, natural disasters. Probably it would help the reader if you reduce the number of dufferent terms you use in the paper.

6. Conclusion "Our study shows that a combination of different methods and approaches is crucial to a comprehensive understanding of interactions between natural hazards, forest ecosystems and human drivers" natural hazards => only snow avalanches and debris flows, which is not that well understood because of the difficulty to separate the two human drivers => where does this suddenly come from, not analysed in the article

"provide a sufficient basis for decision support" decision support on what?

"This is particularly true for complex mountainous regions like the Nevados de Chillán" why are they complex?

"the combination of different methods applied in this study suggests that the conservation of regional native forests may contribute" The findings of the study may suggest that, not the methods.

"it is important to better integrate related spatial information as an input for land-use planning tools" rephrase this sentence please

7. Overall remark: the scope of the paper needs to be improved, a good red line is currently missing. A range of different results are presented and finally linked with Eco-DRR in the discussion without properly explaining what this concretely means in this region.

8. Figure improvement - Fig. 1: Focus is on the yellow square- please zoom in on the yellow square and the red line - Fig. 4: Replace plot number by "forest plot (fp)

number" - Fig. 6 : scale is missing; the green dots are probably sampled trees - not mentioned;

---

## Referee Comment (RC2) · Anonymous Referee #2 · 19 Jan 2018

This is an interesting paper dealing with the assessment of the interaction between mountain forests and natural hazards in Chile, with a special focus on snow avalanches. The paper is well written and organized. A few remarks have been listed below: Title: The title should be more detailed: The papers in fact deals with snow avalanches and to a smaller extent to other natural hazards. Pag 1 Line 3: Change more into most Pag 1 Line 28: I suggest to add snow before sheds Pag 2 Line 5: Do you mean interaction with the people living in mountain areas? Pag 2 Line 37: Change into to better understand the triggers and dynamics of such natural hazards Pag 2 Line

39: Do these services could be considered as ecosystem services? Pag 2 Line 40: Change timer into timber. What kind of food native forests provide? Pag 3 Line 28: Why did you not include the description of the study area and of the biogeographic setting in the Materials and Methods section? Pag 4 Line 12: What is the elevation of the weather station? Pag 4 Line 13: Does this value of precipitation include the snow water equivalent? Do you have data about rainfall intensity in the area? I think this parameter could have an effect of debris flow phenomena. Pag 4 Line 21: Change inceptisols into Inceptisols. Add as a reference: Soil Survey Staff, 2014. Soil Survey Staff: Soil taxonomy, Twelfth Edition, USDA, Washington DC, USA, 2014. Pag 5 Lines 13-16: I think these sentences should be moved to the results section Pag 8 Line 12: Do you mean artificial afforestation? Pag 8 Line 18: Change natural disasters into natural hazards Pag 8 Line 36: Change into reactions wood Pag 9 Line 16: Delete : after RAMMS Pag 9 Line 33: see comment on Pag 1 Line 28 Pag 10 Line 4: Delete . Fig 4: Please in the caption add the meaning of the different colors. Fig 6: Please in the caption specify the meaning of the green points.

---

## Author Comment (AC1) · 29 Jan 2018

Dear Anonymous Referee #1 Thanks for the valuable comments and suggestions for improving our manuscript. As indicated by the Editorial Support of Copernicus Publications, we reply below to your comments so that the Editor can make a decision about the further handling of our manuscript.

1. The title in this paper needs to be more detailed: The papers deals with snow avalanches and to a smaller extent also with debris flows.

[Figure]

Reply: The new suggested title is the following: "Assessing the interaction between mountain forests and snow avalanches at Nevados de Chillán, Chile, and its implications for Ecosystem-based Disaster Risk Reduction".

2. The introduction needs a better structure: The first part deals with event documentation (in addition, the first and second sentence have no link). Then how settlement in mountains other than the Alps increased substantially and "thus it is particularly important in such more remote mountain areas, in which human populations and infrastructure are expanding, to learn more about different natural hazard processes and how they interact with mountain ecosystems". Why that? I would say it is important to know how these processes potentially affect the humans living there.

Reply: The need to learn more about different natural hazard processes and how they interact with mountain ecosystems in places where there is a lack of historical records is aimed at having better hazard and risk assessments. We will clarify this in the manuscript.

Next the introduction continues with how lacking event information can be obtained via dendrochronological methods. Then we pass on to the statement "forests can protect infrastructure from natural hazards in mountainous areas" and after that the authors only mention the interaction between forests and snow avalanches. And the paragraph ends with Eco-DRR, which is shortly introduced.

Reply: We will state in a more clear way that the main focus of our study is snow avalanches.

The goal is the paper is not specific enough: The overreaching goal of this investigation was to evaluate the role of mountain forest ecosystems as an Eco-DRR measure against natural hazards at Valle de Las Trancas, Nevados de Chillán, in the Biobío Region of Chile. The primary focus of the whole paper is on snow avalanches.

Reply: The manuscript was indeed originally broader (considering different natural

hazards), yet in the revised version we will make clear that the main focus of our study is on snow avalanches.

3. Make sure there is a link between the methods chapter and the results chapter. Suggestion for the structure of the methods chapter 1. Introduction - find the relevant tracks in the forest 2. Forest structure analysis 3. Tree ring analysis 4. Avalanche simulation with forest effect.

Reply: In the revised version (also following the advice of Anonymous Referee #2), the description of the study area and of the biogeographic setting will be included in the Materials and Methods chapter. Then we agree that in this chapter we can separate between "Forest structure analysis" and "Tree ring analysis" to make a clearer link with the Results chapter and we will do this structural modification.

4. The methods chapter mentions (p. 6, line 14): "In some of the avalanche tracks, we also observed abundant evidence of the occurrence of debris flows which were con-firmed in some cases by documentary records" and "it was not possible to differentiate between the snow avalanches and debris flows". Then in the results (p.7 /line 18) the authors mentions: "The results of our tree-ring analyses allowed us to reconstruct past snow avalanche and debris flows years" This all confusing. In the discussion again: "In the current investigation area, however, we were not able to distinguish between avalanche and debris flow events using tree-ring methods because both processes occur typically during the same season (winter), when most of the precipitation com-monly occurs. However, through available records, types of damages in the forest, topographical features (including channel geometry) and process modelling we were able to distinguish - to a large extent - one process from the other". The paper needs to provide more clarity!

Reply: We acknowledge that the different parts of the manuscript related to the dis-tinction between the occurrence of debris flows and snow avalanches are currently not clear enough. In the revised version will better describe up to which extent it was

methodologically possible to distinguish one process from the other and make this also clearer in the result and the discussion chapters.

5. The paper refers to the snow avalanches and debris flows as: natural hazards, natural events, disturbances, natural disasters. Probably it would help the reader if you reduce the number of different terms you use in the paper.

Reply: We agree with this suggestion and we will reduce the number of different terms in the revised version. We will thus refer mainly to "natural hazards" when referring to snow avalanches, debris flows or other natural phenomena with potentially direct negative effects on humans. For other natural disturbances in forest ecosystems (with no direct impact on human settlement or infrastructure) we will use the term "natural disturbances".

6. Conclusion "Our study shows that a combination of different methods and approaches is crucial to a comprehensive understanding of interactions between natural hazards, forest ecosystems and human drivers" natural hazards => only snow avalanches and debris flows, which is not that well understood because of the difficulty to separate the two human drivers => where does this suddenly come from, not analysed in the article.

Reply: As indicated in a previous comment, we will clarify in the revised version of the manuscript the way and to which extent it was methodologically possible to distinguish between the occurrence of snow avalanches and debris flows at the study area. Human drivers (related here to activities conducted by local communities in the area, such as wood extraction) are not a focus of our investigations and yet we bring this issue in the discussion and conclusion chapters to show the broader picture.

"provide a sufficient basis for decision support" decision support on what?

Reply: We mean here that all elements need to be considered for a comprehensive decision support about risk reducing measures, such as organizational, related to spatial planning, afforestations and avalanche barriers. We will include this information in the revised version for more clarity.

"This is particularly true for complex mountainous regions like the Nevados de Chillán" why are they complex?

Reply: We mean here that this is a region of occurrence of multiple natural hazards (and other natural disturbances) that sometimes interact with each other (e.g. volcanic activity triggering snow avalanches or mud flows). In addition, the complex topography (steep mountain terrain) results in a more difficult acquisition of accurate digital terrain models which are necessary for the applied simulation models. We will include this new information in the revised manuscript.

"the combination of different methods applied in this study suggests that the conservation of regional native forests may contribute" The findings of the study may suggest that, not the methods.

Reply: We will modify this statement as suggested for more clarity in the revised version.

"it is important to better integrate related spatial information as an input for land-use planning tools" rephrase this sentence please

Reply: We will rephrase this sentence for more clarity in the revised version.

7. Overall remark: the scope of the paper needs to be improved, a good red line is currently missing. A range of different results are presented and finally linked with Eco-DRR in the discussion without properly explaining what this concretely means in this region.

Reply: We will better integrate the concept of Eco-DRR in our study by including other relevant examples and citations from the region (which are in fact quite limited) but also from other regions of the world.

[Figure]

8. Figure improvement - Fig. 1: Focus is on the yellow square- please zoom in on the yellow square and the red line - Fig. 4: Replace plot number by "forest plot (fp) number" - Fig. 6 : scale is missing; the green dots are probably sampled trees – not mentioned;

Reply: Fig. 1: we will improve the figure as suggested. Fig. 4: We will do the indicated replacement. Fig. 6: we will add a scale and indicate what the green dots represent in the figure.
* * *

---

## Author Comment (AC2) · 29 Jan 2018

Dear Anonymous Referee #2 Thanks for the valuable comments and suggestions for improving our manuscript. As indicated by the Editorial Support of Copernicus Publications, we reply below to your comments so that the Editor can make a decision about the further handling of our manuscript.

This is an interesting paper dealing with the assessment of the interaction between mountain forests and natural hazards in Chile, with a special focus on snow

avalanches. The paper is well written and organized. A few remarks have been listed below: Title: The title should be more detailed: The papers in fact deals with snow avalanches and to a smaller extent to other natural hazards.

Reply: The new suggested title is the following: "Assessing the interaction between mountain forests and snow avalanches at Nevados de Chillán, Chile, and its implications for Ecosystem-based Disaster Risk Reduction".

Pag 1 Line 3: Change more into most

Reply: The text will be modified as suggested.

Pag 1 Line 28: I suggest to add snow before sheds

Reply: The text will be modified as suggested.

Pag 2 Line 5: Do you mean interaction with the people living in mountain areas?

Reply: We mean here interactions in mountain regions between people, natural hazards and ecosystems. We will clarify this statement in the revised version.

Pag 2 Line 37: Change into to better understand the triggers and dynamics of such natural hazards

Reply: The text will be modified as suggested.

Pag 2 Line 39: Do these services could be considered as ecosystem services?

Reply: Yes, this is correct. We will include the word "ecosystem" in the indicated sentence in the revised version.

Pag 2 Line 40: Change timer into timber. What kind of food native forests provide?

Reply: The text will be modified as suggested. Araucaria araucana forests have been for centuries an important provider of food (seeds) for local populations. We will briefly mention this in the revised version.

Pag 3 Line 28: Why did you not include the description of the study area and of the biogeographic setting in the Materials and Methods section?

Reply: We will restructure the manuscript in order to include the description of the study area and of the biogeographic setting in the Materials and Methods section.

Pag 4 Line 12: What is the elevation of the weather station?

Reply: The elevation of the weather station is.1251 m asl. We will include this information in the revised version.

Pag 4 Line 13: Does this value of precipitation include the snow water equivalent? Do you have data about rainfall intensity in the area? I think this parameter could have an effect of debris flow phenomena.

Reply: The provided precipitation value does not include the snow water equivalent. Unfortunately, we do not have data about rainfall intensity in the area. We will mention this in the revised version of manuscript.

Pag 4 Line 21: Change inceptisols into Inceptisols. Add as a reference: Soil Survey Staff, 2014. Soil Survey Staff: Soil taxonomy, Twelfth Edition, USDA, Washington DC, USA, 2014.

Reply: The text will be modified as suggested and the new reference included in the revised version.

Pag 5 Lines 13-16: I think these sentences should be moved to the results section

Reply: We agree that these two sentences would fit better in the results section and we will do the requested change in the revised version.

Pag 8 Line 12: Do you mean artificial afforestation?

Reply: Yes, we will clarify this point in the revised version of the manuscript.

Pag 8 Line 18: Change natural disasters into natural hazards

Reply: The text will be modified as suggested (see also related comment and reply for Anonymous Referee #1).

Pag 8 Line 36: Change into reactions wood

Reply: The text will be modified as suggested.

Pag 9 Line 16: Delete : after RAMMS

Reply: The way the name of the program was indicated is correct (for your reference, see: http://ramms.slf.ch/ramms/index.php?option=com_content&view=article&id=60&Itemid=77).

Pag 9 Line 33: see comment on Pag 1 Line 28

Reply: The text will be modified as suggested.

Pag 10 Line 4: Delete .

Reply: The text will be modified as suggested.

Fig 4: Please in the caption add the meaning of the different colors.

Reply: the meaning of the different colours is already indicated in the original figure caption.

Fig 6: Please in the caption specify the meaning of the green points.

Reply: The figure will be modified as suggested.

---

## Author Response (AR1)

Dear Dr. Thom Bogaard, editor

Please find attached a revised version of our manuscript (nhess-2017-348) newly entitled "Assessing the interaction between mountain forests and snow avalanches at Nevados de Chillán, Chile, and its implications for Ecosystem-based Disaster Risk Reduction".

We hope to have answered thoroughly all the valuable comments and suggestions of the two anonymous referees.

With best regards

Dr. Alejandro Casteller & co-authors

REPLIES TO REFEREES' COMMENTS

**Dear Anonymous Referee #1**

Thanks for the valuable comments and suggestions for improving our manuscript. As indicated by the Editorial Support of Copernicus Publications, we reply below to your comments so that the Editor can make a decision about the further handling of our manuscript.

1. The title in this paper needs to be more detailed: The papers deals with snow avalanches and to a smaller extent also with debris flows

**Reply**: The title has been modified as follows: "Assessing the interaction between mountain forests and snow avalanches at Nevados de Chillán, Chile, and its implications for Ecosystem-based Disaster Risk Reduction".

2. The introduction needs a better structure:

The first part deals with event documentation (in addition, the first and second sentence have no link). Then how settlement in mountains other than the Alps increased substantially and "thus it is particularly important in such more remote mountain areas, in which human populations and infrastructure are expanding, to learn more about different natural hazard processes and how they interact with mountain ecosystems".

Why that? I would say it is important to know how these processes potentially affect the humans living there.

**Reply**: The first and second sentences have been better linked (P 1, lines 31-35, revised version). The need to learn more about different natural hazard processes and how they interact with mountain ecosystems in places where there is a lack of historical records is aimed at having better hazard and risk assessments. We have clarified this point in the manuscript (P. 2, lines 4-8, revised version; see related comment from Referee #2).

Next the introduction continues with how lacking event information can be obtained via dendrochronological methods.

Then we pass on to the statement "forests can protect infrastructure from natural hazards in mountainous areas" and after that the authors only mention the interaction between forests and snow avalanches. And the paragraph ends with Eco-DRR, which is shortly introduced.

**Reply**: We have stated in a more clear way that the main focus of our study is snow avalanches (P. 2, lines 21-23, revised version).

The goal is the paper is not specific enough: The overreaching goal of this investigation was to evaluate the role of mountain forest ecosystems as an Eco-DRR measure against natural hazards at Valle de Las Trancas, Nevados de Chillán, in the Biobío Region of Chile. The primary focus of the whole paper is on snow avalanches.

**Reply**: The manuscript was indeed originally broader (considering different natural hazards), yet in the revised version we have made clear that the main focus of our study is on snow avalanches (P. 3, lines 16-19, revised version).

3. Make sure there is a link between the methods chapter and the results chapter. Suggestion for the structure of the methods chapter 1. Introduction - find the relevant tracks in the forest 2. Forest structure analysis 3. Tree ring analysis 4. Avalanche simulation with forest effect.

**Reply**: In the revised version (also following the advice of Anonymous Referee #2), the description of the study area and of the biogeographic setting have been included in the Materials and Methods chapter. Further, we have separated in the revised version between "Forest structure analysis" and "Tree-ring analyses" to make a clearer link with the Results chapter.

4. The methods chapter mentions (p. 6, line 14): "In some of the avalanche tracks, we also observed abundant evidence of the occurrence of debris flows which were confirmed in some cases by documentary records" and "it was not possible to differentiate between the snow avalanches and debris flows". Then in the results (p.7 /line 18) the authors mentions: "The results of our tree-ring analyses allowed us to reconstruct past snow avalanche and debris flows years" This all confusing. In the discussion again: "In the current investigation area, however, we were not able to distinguish between avalanche and debris flow events using tree-ring methods because both processes occur typically during the same season (winter), when most of the precipitation commonly occurs. However, through available records, types of damages in the forest, topographical features (including channel geometry) and process modelling we were able to distinguish - to a large extent - one process from the other". The paper needs to provide more clarity!

**Reply**: We acknowledge that the different parts of the manuscript related to the distinction between the occurrence of debris flows and snow avalanches were not clear enough. In the revised version, we have clarified the methods used to distinguish one process from the other (P. 6, lines 25-26; P. 9, lines 13-19, revised version).

5. The paper refers to the snow avalanches and debris flows as: natural hazards, natural events, disturbances, natural disasters. Probably it would help the reader if you reduce the number of different terms you use in the paper.

**Reply**: We agree with this suggestion and we have reduced the number of different terms in the revised version. We have stated mainly to "natural hazards" when referring to snow avalanches, debris flows or other natural phenomena with potentially direct negative effects on humans. For other natural disturbances in forest ecosystems (with no direct impact on human settlement or infrastructure) we have used the term "natural disturbances".

6. Conclusion "Our study shows that a combination of different methods and approaches is crucial to a comprehensive understanding of interactions between natural hazards, forest ecosystems and human drivers" natural hazards => only snow avalanches and debris flows, which is not that well understood because of the difficulty to separate the two human drivers => where does this suddenly come from, not analysed in the article

**Reply**: As indicated in a previous comment, we have clarified in the revised version of the manuscript the way and to which extent it was methodologically possible to distinguish between the occurrence of snow avalanches and debris flows at the study area. Human drivers

(related here to activities conducted by local communities in the area, such as wood extraction) are not a focus of our investigations and yet we bring this issue in the discussion and conclusion chapters to show the broader picture. In any case, we have reformulated the first sentence of the conclusion to make our statement clearer (P. 10, lines 25-27, revised version).

"provide a sufficient basis for decision support" decision support on what?

**Reply**: We mean here that all elements need to be considered for a comprehensive decision support about risk reducing measures, such as organizational, related to spatial planning, afforestations and avalanche barriers. We reformulated this sentence to be clearer (P. 10, lines 25-27, revised version).

"This is particularly true for complex mountainous regions like the Nevados de Chillán" why are they complex?

**Reply**: We mean here that this is a region of occurrence of multiple natural hazards (and other natural disturbances) which sometimes interact with each other (e.g. volcanic activity triggering snow avalanches or mud flows). In addition, the complex topography (steep mountain terrain) results in a more difficult acquisition of accurate digital terrain models which are necessary for the applied simulation models. We have included this new information in the revised manuscript (P. 10, lines 27-31, revised version).

"the combination of different methods applied in this study suggests that the conservation of regional native forests may contribute" The findings of the study may suggest that, not the methods.

**Reply**: We have modified this statement as suggested for more clarity in the revised version (P. 10, lines 31-34, revised version).

"it is important to better integrate related spatial information as an input for land-use planning tools" rephrase this sentence please

**Reply**: We have rephrased this sentence for more clarity in the revised version (P. 10, lines 34-36, revised version).

7. Overall remark: the scope of the paper needs to be improved, a good red line is currently missing. A range of different results are presented and finally linked with Eco-DRR in the discussion without properly explaining what this concretely means in this region.

**Reply**: We have better integrated the concept of Eco-DRR in our study by including other relevant examples and citations from the study region and also from other regions of the world (P. 2, lines 35-40, revised version).

8. Figure improvement - Fig. 1: Focus is on the yellow square- please zoom in on the yellow square and the red line - Fig. 4: Replace plot number by "forest plot (fp) number" - Fig. 6 : scale is missing; the green dots are probably sampled trees – not mentioned;

**Reply**: Fig. 1: we have redone the figure as suggested. Fig. 4: we have redone the figure as suggested. Fig. 6: we have added a scale and indicated what the green dots represent in the figure.
* * *
**Dear Anonymous Referee #2**

Thanks for the valuable comments and suggestions for improving our manuscript. As indicated by the Editorial Support of Copernicus Publications, we reply below to your comments so that the Editor can make a decision about the further handling of our manuscript.

This is an interesting paper dealing with the assessment of the interaction between mountain forests and natural hazards in Chile, with a special focus on snow avalanches. The paper is well written and organized. A few remarks have been listed below:

Title: The title should be more detailed: The papers in fact deals with snow avalanches and to a smaller extent to other natural hazards.

**Reply**: The new title is the following: "Assessing the interaction between mountain forests and snow avalanches at Nevados de Chillán, Chile, and its implications for Ecosystem-based Disaster Risk Reduction".

Pag 1 Line 3: Change more into most

**Reply**: The text has been modified as suggested (P. 1, line 22, revised version).

Pag 1 Line 28: I suggest to add snow before sheds

**Reply**: The text has been modified as suggested (P. 1, line 29, revised version).

Pag 2 Line 5: Do you mean interaction with the people living in mountain areas?

**Reply**: We mean here interactions in mountain regions between people, natural hazards and ecosystems. This statement has been clarified in the revised version (P. 2, lines 4-8, revised version).

Pag 2 Line 37: Change into to better understand the triggers and dynamics of such natural hazards

**Reply**: The text has been modified as suggested in the revised version (P. 3, lines 4-7, revised version).

Pag 2 Line 39: Do these services could be considered as ecosystem services?

**Reply**: Yes, this is correct. We have included the word "ecosystem" in the indicated sentence in the revised version (P. 3, line 8, revised version).

Pag 2 Line 40: Change timer into timber. What kind of food native forests provide?

**Reply**: The text has been modified as suggested. Araucaria araucana forests have been for centuries an important provider of food (seeds) for local populations. We have briefly mentioned this in the revised version (P. 3, lines 8-11, revised version).

Pag 3 Line 28: Why did you not include the description of the study area and of the biogeographic setting in the Materials and Methods section?

**Reply**: We have restructured the manuscript in order to include the description of the study area and of the biogeographic setting in the Materials and Methods section.

Pag 4 Line 12: What is the elevation of the weather station?

**Reply**: The elevation of the weather station is.1251 m asl. We have included this information in the revised version (P. 4, line 24, revised version).

Pag 4 Line 13: Does this value of precipitation include the snow water equivalent? Do you have data about rainfall intensity in the area? I think this parameter could have an effect of debris flow phenomena.

**Reply**: The provided precipitation value does not include the snow water equivalent. Unfortunately, we do not have data about rainfall intensity in the area. We have mentioned this in the revised version of manuscript (P. 4, lines 25-27, revised version).

Pag 4 Line 21: Change inceptisols into Inceptisols. Add as a reference: Soil Survey Staff, 2014. Soil Survey Staff: Soil taxonomy, Twelfth Edition, USDA, Washington DC, USA, 2014.

**Reply**: The text has been modified as suggested and the new reference included in the revised version (P. 4, lines 32-34, revised version).

Pag 5 Lines 13-16: I think these sentences should be moved to the results section

**Reply**: Since some of the information provided here is needed for the following point (Forest structural and dendrochronological analyses), we had to preserve here some content of the original text (P. 5, lines 23-25, revised version).

Pag 8 Line 12: Do you mean artificial afforestation?

**Reply**: Yes, we have clarified this point in the revised version of the manuscript (P. 8, line 22, revised version).

Pag 8 Line 18: Change natural disasters into natural hazards

**Reply**: The text has been modified as suggested in the revised version (P. 8, line 28, revised version; see also related comment and reply for Anonymous Referee #1).

Pag 8 Line 36: Change into reactions wood

**Reply**: The text has been modified as suggested in the revised version (P. 9, line 10, revised version).

Pag 9 Line 16: Delete : after RAMMS

**Reply**: The way the name of the program was indicated is correct (for your reference, see: http://ramms.slf.ch/ramms/index.php?option=com_content&view=article&id=60&Itemid=77; P. 9, line 30, revised version).

Pag 9 Line 33: see comment on Pag 1 Line 28

**Reply**: The text has been modified as suggested (P. 10, line 10, revised version).

Pag 10 Line 4: Delete .

**Reply**: The text has been modified as suggested (P. 10, line 19, revised version).

Fig 4: Please in the caption add the meaning of the different colors.

**Reply**: The meaning of the different colours is already indicated in the original figure caption.

Fig 6: Please in the caption specify the meaning of the green points.

**Reply**: The figure caption has been modified as suggested.

[revised manuscript text omitted]

---

## Author Response (AR2)

Dear Dr. Thom Bogaard, editor

Please find attached a revised version of our manuscript (nhess-2017-348) entitled "Assessing the interaction between mountain forests and snow avalanches at Nevados de Chillán, Chile, and its implications for Ecosystem-based Disaster Risk Reduction".

I this version of the manuscript, we have reformulated the conclusions in order to address all the research questions raised in the introduction.

With best regards

Dr. Alejandro Casteller & co-authors

[revised manuscript text omitted]

---

## Author Response (AR3)

Dear Dr. Thom Bogaard, editor

Please find attached a revised version of our manuscript (nhess-2017-348) entitled "Assessing the interaction between mountain forests and snow avalanches at Nevados de Chillán, Chile, and its implications for Ecosystem-based Disaster Risk Reduction".

I this version of the manuscript, we have addressed the previously suggested changes of referee #2, which we did not note earlier.

With best regards

Dr. Alejandro Casteller & co-authors

Dear Anonymous Referee #2

Thanks for the valuable additional comments and suggestions for improving our manuscript. Please find a reply to your comments below:

The paper has considerably improved after the first revision and I think that now it's ready for publication in the journal.

A few more things are reported below:

Pag 11 Line 25: After 1007 mm add (excluding the snow water equivalent)

**Reply:** the information has been included (P. 4, lines 21-22, revised version).

Pag 13 Line 24: Delete during

**Reply:** the word during has been deleted (P. 6, line 21, revised version).

Pag 14 Line 6: How did you collect these informations?

**Reply:** the information regarding snow height and snow density along the track and at the release areas had to be estimated for some events based on typical and extreme snow situations for this region. For this purpose we took contact with snow and avalanche experts (including locals) who provided us the guidelines. In addition, the documented events served as a basis to extrapolate to other events with less available information.

Pag 14 Line 13: Change into Feistl et al. (2015, 2016)

**Reply:** the correction has been made (P. 7, lines 7-8, revised version).

Pag 15 Line 22: Is artificial afforestation a commone practice in the area? I was thinking that this practice should be described in the Introduction section.

**Reply:** the artificial afforestations are scenarios, not a common practice in the area. As we consider this is clearly stated in the manuscript, we have not modified the original text.

Pag 15 Line 28: Change landislides into debris flows. See also pag 16, line 2.

**Reply:** the word substitution has been made (P. 8, line 23 and P. 8, line 33, revised version)

Pag 15 Line 35: What about the artificial afforestation? See also Pag 17 Line 9

**Reply:** as indicated in a previous comment, the artificial afforestations are scenarios and consider the same type of forests as the ones growing in the area. Therefore we consider it not necessary to modify the text in its current form and content.

Pag 17 Line 13: Delete a space after (….2017)

**Reply:** the correction has been made (P. 10, line 3, revised version)

[revised manuscript text omitted]